# NMR-Based Metabolomic Profiling for Brain Cancer Diagnosis and Treatment Guidance

**DOI:** 10.3390/metabo15090607

**Published:** 2025-09-11

**Authors:** Julia R. Zickus, José S. Enriquez, Paytience Smith, Bill T. Sun, Muxin Wang, Aldo Morales, Pratip K. Bhattacharya, Shivanand Pudakalakatti

**Affiliations:** 1Department of Cancer Systems Imaging, University of Texas MD Anderson Cancer Center, Houston, TX 77054, USA; jrzickus@mdanderson.org (J.R.Z.); jsenriquez@mdanderson.org (J.S.E.); smithpaytience11@gmail.com (P.S.); btsun@mdanderson.org (B.T.S.); mwang15@mdanderson.org (M.W.); atmorales@mdanderson.org (A.M.); pkbhattacharya@mdanderson.org (P.K.B.); 2Graduate School of Biomedical Sciences, University of Texas MD Anderson Cancer Center UT Health Houston, Houston, TX 77030, USA

**Keywords:** NMR spectroscopy, brain cancer, metabolomics, diagnosis, therapeutic intervention

## Abstract

Nuclear magnetic resonance (NMR) spectroscopy is a routinely used analytical tool for studying chemical entities of varying molecular sizes, ranging from approximately 20 Da to ~45 kDa, and in some cases even larger. Over the past two decades, the use of NMR spectroscopy has significantly expanded to the study of metabolomics. In this medium-sized review, the application of NMR-based metabolomics in the diagnosis, therapeutic intervention, and guidance of therapy for various types of brain cancer is discussed.

## 1. Background: Nuclear Magnetic Resonance (NMR) Spectroscopy

Nuclear magnetic resonance (NMR) spectroscopy is a widely applied analytical tool to study small organic molecules to larger biopolymers like proteins, RNA, and DNA. One of the major applications of NMR spectroscopy in the past two decades has been in the field of metabolomics. Metabolomics is the study of small molecules of <2000 Da, known as metabolites, within cells, biofluids, or tissues. This technique can describe the underlying biochemical activity and state of cells to represent the molecular phenotype [1]. NMR spectroscopy can identify numerous metabolites in complex mixtures based on their chemical structure. Additional analytical information can also be provided by NMR due to spin-spin coupling, whereby resonances are split into multiplets because of through-bond interactions between nuclear spins [2]. Unlike other metabolomic techniques, NMR spectroscopy is highly reproducible and quantitative, without the need for intense sample preparation. NMR is also a non-destructive technique that is easy to analyze and quantify. It can detect a wide range of compounds, whether in pure form or within mixtures, making it an ideal analytical tool for metabolomics studies [3]. NMR can also assess non-polar soluble molecular entities such as lipids when combined with Gas Chromatography and/or liquid chromatography. This review comprehensively discusses the application of NMR spectroscopy in the early diagnosis and therapeutic intervention of brain cancer.

## 2. Metabolism in Brain Cancer

The brain relies heavily on glucose as its main substrate for energy metabolism. Unlike other organs, which can utilize lipids, amino acids, or other sugars as a source of energy or operate under hypoxic conditions briefly, the brain is exclusively dependent on blood glucose and is responsible for a quarter of total glucose consumption in the body. The brain requires high energy consumption to fuel metabolic processes, including the operation of ion pumps, the synthesis of proteins, and the stabilization of the membrane [4]. Glucose enters cells through glucose transporters (GLUTS) and is converted into glucose-6-phosphate by hexokinase. Glucose-6-phosphate is mainly produced through glycolysis as an intermediate to mitochondrial metabolism, but it can be produced through the pentose phosphate pathway, or glycogenesis in astrocytes [5]. In optimal conditions, glucose is the sole substrate for the brain. However, under certain conditions, the brain has the capacity to adapt and utilize other energy substrates such as ketone bodies or lactate [6,7,8]. Some studies have also shown that specific neural cells can adapt to keep metabolic homeostasis in the brain. When neuronal activity is high, neurotransmitters such as potassium and glutamate are released into the extracellular space. This increase triggers astrocyte hyperglycolysis to maintain the chemical balance in the cortical microenvironment. The increased glycolytic activity from the astrocytes provides neurons with ample extracellular lactate to use as an energy source [9].

Cancer cells alter metabolism as a determinant of cell proliferation. To rapidly grow and divide, cancer cells must produce cellular energy to replicate their cellular contents. Rather than depend on the TCA cycle to produce ATP, in most cancer types, cancer cells preferentially rely on the conversion of glucose to lactate, irrespective of oxygen availability, defined as the Warburg effect. This is due to the inhibition of pyruvate dehydrogenase (PDH) activity by pyruvate dehydrogenase kinase 1 and an increase in lactate dehydrogenase A (LDH) activity [10]. In this review, we discuss the altered metabolic pathways observed in various subtypes of brain cancer, as studied through NMR spectroscopy, with a focus on their relevance to diagnosis, prognosis, and therapeutic intervention. Additionally, we refer readers to other comprehensive reviews that explore brain cancer metabolism in humans and animal models across in vitro, ex vivo, and in vivo imaging contexts. These studies employ a range of analytical techniques, including NMR spectroscopy, magnetic resonance spectroscopy (MRS), hyperpolarized MRS, and mass spectrometry, to advance understanding of metabolic alterations for diagnostic and prognostic applications [11,12,13].

### 2.1. Gliomas

The World Health Organization (WHO) revised its brain tumor classification to use a layered diagnosis concept, which combined histology and molecular pathology twice in the last decade (2016 and 2021) [14]. Gliomas are classified into three broad groups depending on the deletion status of chromosomal arms 1p and 12q, and the mutation status of the citric acid (TCA) cycle enzyme isocitrate dehydrogenase 1 (IDH1) or the mitochondrial IDH2 [15,16]. Glioblastomas are classified as those that are IDH wildtype, and astrocytomas and oligodendrogliomas have the IDH1 mutation, but oligodendrogliomas are further defined by a 1p19q chromosomal codeletion [16].

#### 2.1.1. Glioblastoma

Glioblastoma multiforme (GBM) is a grade IV astrocytoma and the most common malignant brain tumor in adults, representing almost half of all malignant primary brain tumors (46.1%) [17]. Neural stem and progenitor cells (NSPC) retain the ability to replicate and have been suggested as the most likely cells of origin for glioblastoma. Immunohistochemical markers often support glioblastoma diagnosis, including glial fibrillary acidic protein expression to verify differentiation into the astrocyte lineage. Glioblastomas create a hypoxic tumor environment due to their rapid proliferation, causing the healthy cells in these low-oxygen environments to die or undergo necrosis. The tumor cells will then migrate away from the necrotic center, creating a dense band of tumor cells in a palisade-like pattern around the dead tissues. This is called pseudopalisading necrosis and is one of the WHO criteria for diagnosing glioblastoma [18].

Due to the rapid proliferation and hypoxic environment created by these cancer cells, it is not surprising that there is a metabolic shift observed in glioblastoma. An in vitro study of nine glioblastoma cell lines found that metabolic data can be used to distinguish between glioblastoma cell types [19]. The nine cell lines they used were clustered in four districted groups based on principal components, the fuzzy K-means method, and the hierarchical clustering method. Group 1 cell lines include LN229 and LN219, which showed significantly higher choline than all other samples, as well as the overconcentration of inositol and myo-inositol. When comparing with the other three groups, they found that taurine, uridine diphosphate, choline, phosphocholine (PC), glycerophosphocholine (GPC), glycine, myo-inositol, glutamine, glutamate, citric acid, aspartate, asparagine, and methionine are the most significantly upregulated metabolites. At the transcription level, these cell lines strongly express PDGFRA but have low levels of EGFR [19].

The group 2 cell lines, HS683 and LN405, saw a reduced concentration of glutamine, which is consistent with the overexpression of the OGDH gene. Valine, leucine, isoleucine, alanine, lactic acid, glutamate, citric acid, aspartate, asparagine, and methionine are the most significantly upregulated metabolites in group 2 cell lines compared to the other group 1, 3, and 4 cell lines. Upregulation of glutamate and aspartate was observed in the group 3 cell lines, A172, U343, and LN18. GABA, proline, glutamine, glutamate, methionine, citric acid, aspartate, and asparagine were found to be the most upregulated metabolites in comparison to the other groups. Glutamine and glutamate were found to be downregulated in the group 4 cell lines, U373 and BS149, whereas upregulation was observed in glycerol 3-phosphate (G3P). The most significantly upregulated metabolites in comparison to the other groups included succinic acid, serine, adenine, taurine, lysine, tyrosine, G3P, glucose, cis-aconitic acid, GABA, and proline [19]. This study was able to connect metabolomics results with gene expression analysis, which concluded that different GBM cell lines had different metabolic profiles. However, to further support these results, similar experiments performed with clinical samples would be more representative of the patient tumors.

A different NMR study employed human urine and blood samples from 70 glioma patients and 70 healthy individuals. They found 20 metabolites to have differential levels, including citrate, pyruvate, glucose, formate, and creatine, which can be used to distinguish plasma samples from healthy individuals and those with tumors. Increased levels of phospholipids and total cholesterol were observed in the plasma profiles of brain tumor patients in comparison to healthy individuals [20]. The large sample size used in this study is an advantage compared to many other reported NMR spectroscopic studies. This was the first NMR study that demonstrated metabolites in plasma can help distinguish between a healthy individual from a patient with glioma.

In another study focused on platelet metabolism in brain cancer patients—including those with GBM, astrocytoma, medulloblastoma, gliosarcoma, and meningioma—altered metabolic profiles were observed compared to platelets from healthy volunteers. In general, the concentrations of metabolites such as lactate, acetate, glutamine, glutamate, succinate, alanine, and pyruvate were decreased in patient samples. This represents one of the first pilot studies to explore platelet-based metabolism to distinguish brain cancer patients from healthy individuals using NMR spectroscopy. However, the study was conducted on a small sample size, necessitating further validation with larger cohorts. Nonetheless, the findings are intriguing and warrant continued investigation [21].

Gliosarcoma is a rare form of glioblastoma that occurs in the glial cells of the brain surface or spinal cord and is characterized by its distinctive mesenchymal and glial differentiation [22]. Gliosarcoma shows similar characteristics to other glioblastomas and tumors that have metastasized to the brain from other parts of the body when using traditional MRI for diagnosis. NMR spectroscopy has the ability to identify metabolites present in brain tumors, making it a potentially useful tool to better understand the metabolic mechanisms that underlie gliosarcoma tumor growth and treatment response, as well as differentiate between gliosarcoma and other glioblastomas or metastases [23].

NMR can also be utilized to distinguish responsiveness to therapy, as seen in a 2019 NMR study on GBM cultures established from biopsies of 5 patients. In untreated cell lines, citric acid was undetectable; however, when treated with the drug YM155, cells less sensitive to this drug experienced a significant increase in citric acid. The less sensitive cells also saw a decrease in alanine and lactate after treatment [24]. This study demonstrates the usefulness of NMR in cancer treatment evaluation and understanding the efficacy of treatment on specific patients.

A study published in 2021 also explored different metabolic shifts throughout numerous tumor stages in glioblastoma, including during development, regression, and relapse in a mouse model. Throughout tumor development, alanine and PC were the first metabolites to significantly increase in the GBM models compared to controls, followed by glycerolphosphocholine, glycine, and valine later in development. Following radiotherapy, alanine, glycine, and valine were significantly decreased when compared to the untreated tumors. As treatment continued, NAD+ and PC also became significantly decreased in the treated tumors [25] (Figure 1). This study displayed the ability of NMR spectroscopy to track the metabolic profiles of tumors throughout development and treatment and how the tumor metabolism shifts as it continues to grow, as well as how it shifts when responding to treatment. This technique can assist in treatment regimens for patients for evaluating treatments non-invasively.

#### 2.1.2. Astrocytoma

Astrocytomas are defined by a mutation in the isocitrate dehydrogenase 1 (IDH1) gene. This mutation causes the production of 2-HG, which ultimately causes astrocytoma [26]. An IDH1 mutation is caused by a point mutation R132H, which alters the normal function of this enzyme. Rather than catalyzing the oxidative decarboxylation of isocitrate to α-ketoglutarate (α-KG) and producing the cofactor NADPH, this point mutation creates a neomorphic function, converting α-KG into 2-hydroxyglutarate (2-HG) [27] (Figure 2). Although much less common in brain cancer, the mitochondrial IDH2 has various point mutations that can occur, resulting in the same altered function [28]. The most common IDH2 point mutations include a R140Q or R172K point mutation [29]. Tumors that were diagnosed as IDH-mutated glioblastomas prior to 2021 are now classified as astrocytoma IDH-mutated.

An IDH1 mutation is caused by a point mutation, which distinguishes glioblastoma from astrocytoma and oligodendroglioma. This IDH1 mutation produces 2-hydroxyglutarate from a-ketoglutaric acid.

A study published in the Journal of Clinical Neuroscience used human brain tumor samples and demonstrated that elevated 2-HG could be used as a biomarker of the IDH1/2 mutation status in gliomas, which would distinguish glioblastomas from astrocytomas and oligodendrogliomas [30]. The IDH1 mutation is known to silence the glutamine-dependent pathway for metabolites and fatty acids by converting α-KG into 2-HG (Figure 2) [31]. Myo-inositol, an activator of protein C kinase and downstream of α-KG in the TCA cycle, was found to be significantly lower in glioblastoma as compared to astrocytoma and oligodendroglioma [30].

**Figure 2 metabolites-15-00607-f002:**
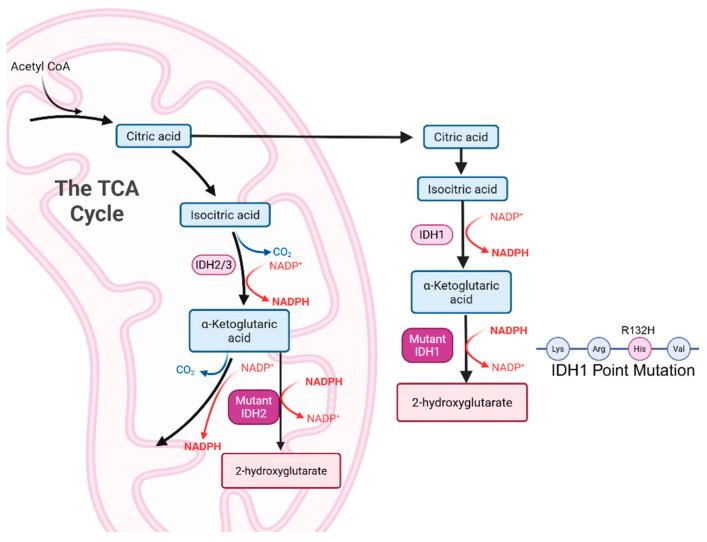
IDH1/2 Point Mutation Produces 2-hydroxyglutarate (2HG), bypassing the TCA Cycle. TCA cycle metabolites are highlighted in rectangular light blue, TCA cycle byproduct 2HG in pink boxes, co-factors are shown in red font, enzymes in pink oval boxes, and mutated enzyme in purple oval boxes. The IDH1 point mutation is highlighted in a pink circular box.

Early NMR studies on astrocytomas found that the absolute choline concentration in astrocytomas in vivo did not differ from normal white matter. However, in a 2010 study of pediatric brain tumors, it was observed that pilocytic astrocytoma had higher concentrations of fatty acids, representing a possible alteration in lipid metabolism in this pathology. Pilocytic astrocytoma is a glial, low-grade tumor with a good prognosis that begins in the astrocyte cells of the central nervous system [32,33]. Separating pilocytic astrocytoma from low-grade diffuse astrocytoma during diagnosis has been difficult and relies heavily on demographic as well as imaging information. This difficulty in differentiation is exacerbated by typically small tissue samples. Consequently, NMR analysis of metabolites could provide another avenue to improve our diagnostic proficiency of pilocytic astrocytoma.

Pilocytic astrocytomas tend to be less aggressive tumors, which was further supported by the low levels of all choline-containing compounds, including PC and GPC [34], as well as low levels of creatine and myo-inositol seen in these tumors compared to medulloblastomas and ependymomas [34]. Higher concentrations of some amino acids, including leucine, isoleucine, and valine, were detected in this study, as well as slightly higher levels of N-acetylaspartate (NAA). NAA being present at higher concentrations may be indicative of less neuronal destruction. A 2018 study using the High Resolution Magic Angle Spinning (HR-MAS) NMR technique found high glutamine and hypotaurine in pilocytic astrocytoma compared to medulloblastoma and ependymoma. Additionally, glutamine, hypotaurine, acetate, NAA, and scyllo-inositol were found to be identifiers of pilocytic astrocytoma [34]. NMR is not only a useful tool to develop a metabolic profile for cancer types and distinguish between different tumor types, but it has also been shown to be useful in distinguishing between different grades and histological differences between tumors, as seen in astrocytomas.

#### 2.1.3. Oligodendroglioma

Oligodendrogliomas are defined by two genetic alterations, like astrocytomas, they have an IDH mutation, and then are further defined by a 1p19q codeletion on the tumor cells’ chromosomes [35].

In a 2008 human tumor study, a metabolic distinction between high-grade and low-grade oligodendrogliomas was established using high-resolution magic angle spinning (MAS) NMR on a 500 MHz spectrometer. At this time, low-grade and high-grade oligodendrogliomas were purely defined by histological features. The greatest metabolic distinction between high and low-grade tumors was observed in the metabolites related to amino acid metabolism. In high-grade oligodendrogliomas, an increase in alanine and valine production was observed, whereas metabolites involved in the Krebs cycle were seen to be decreased, including glutamate, glutamine, GABA, and NAA. A decrease in proline was also observed. This reflects the metabolic shift towards fermentative metabolism, supporting a greater hypoxic tumor environment in the high-grade oligodendrogliomas [36]. However, the sample sizes used had high disparities, with 24 high-grade oligodendrogliomas but only 10 low-grade oligodendrogliomas. The sample sizes also had more samples from men than women, with wide age ranges.

#### 2.1.4. Ependymoma

Ependymomas are one of the most common solid brain tumors found in children [34]. Cerebellar ependymoma is a glial malignant or benign tumor that begins in the radial glial ependymal cells of the cerebellum. Categorization of ependymoma through histological studies has correlated poorly with disease prognosis outcomes, indicating that ependymomas with similar tissue classifications may have significantly different clinical trajectories. Hence, the addition of NMR spectroscopy for the classification and diagnosis of ependymomas based on metabolic data could allow for more accurate prediction of disease outcomes compared to traditional histological methods [37]. 

In a 2010 study, ependymomas were characterized by NMR as having upregulated myo-inositol signals [34]. Myo-inositol is known to be involved in the activation of protein kinase C. These researchers also found NAA levels to be comparable to what they saw in medulloblastoma, at very low concentrations. Compared to medulloblastoma, slightly lower levels of GABA, PC, and GPC have been observed [34]. Despite the 5-tumor sample size, this was the largest NMR study on ependymomas at the time.

A study in 2018 was able to validate these earlier results [34] with an expanded sample size [38]. The authors employed 18 ependymoma human tissue samples and found them to show significantly higher concentrations of myo-inositol and glutathione, while having significantly lower concentrations of leucine when compared with medulloblastoma and pilocytic astrocytomas. This study was conducted using HR-MAS, and the analysis included linear discriminant analysis. Ependymomas were separated from medulloblastoma and pilocytic astrocytomas by the second discriminant function, which found myo-inositol, GPC, glucose, and alanine to be defining metabolites for their classification [38]. Therapeutic protocols vary significantly between pilocytic astrocytomas, ependymomas, medulloblastomas, and ATRT. Being able to differentiate between these diseases using HR-MAS can decrease the time required to form reliable diagnoses, which guide treatment approaches.

### 2.2. Medulloblastoma

As a grade IV tumor, medulloblastomas are one of the most frequent and aggressive metastatic pediatric tumors that affect the central nervous system [34]. The medulloblastoma can also be divided into four molecular subgroups—wingless-related integrated site, sonic hedgehog, group III, and group IV—each having varying prognoses and genetic/clinical factors. The pathways involved in Group III and IV tumors are still undefined, which is of concern as these tumors currently have the worst prognosis. The prognosis, treatment, and survival rates depend on many factors, and there is a risk of negative neurologic side effects after surgical resection that impacts the quality of life in these children [39]. The identification of significant metabolites in this cancer using NMR spectroscopy can support driving precise therapeutic interventions that limit the possibilities of unnecessary and possibly harmful treatments.

Recent studies involving the utilization of NMR metabolomics on medulloblastoma have produced consistent results regarding significant metabolites that are upregulated, such as taurine, myo-inositol, free choline, PC, and GPC. However, as solution state NMR spectroscopy requires large amounts of tissue (~50 mg) and the extraction methods are laborious. HR-MAS is believed to be an advantageous alternative as it requires less sample and has a shorter acquisition time. In an early study, HR-MAS was performed on human tumors of varying grades for the comparison of various pediatric brain cancer metabolic profiles. Results revealed that the five medulloblastomas studied averaged higher levels of taurine, myo-inositol, PC, and GPC when compared to tumors of other grades in a cohort of 30 [40]. Also, when compared to the twelve grade II and eight grade III astrocytomas, as well as the three grade IV glioblastomas studied, medulloblastomas averaged lower levels of creatine and lactate. With this data, the researchers were then able to create a machine-based learning tool that was able to predict tumor grades with 87% accuracy and sensitivity, and 93% specificity [40]. This study demonstrates that HR-MAS spectroscopy could be helpful in accurately determining and classifying tumor grades in medulloblastomas. In another study involving identifying metabolic tumors in pediatric brain tumors, HR-MAS was used to characterize seven medulloblastomas. These tumors exhibited high levels of taurine, myo-inositol, GPC, PC, and choline when compared against a cohort of three grade II and two grade III ependymomas, as well as eight grade I astrocytomas. This study also identified lower levels of fatty acids, creatine, NAA, and glucose in medulloblastomas [34]. The data in this study indicate that metabolic profiling could be utilized to assess brain tumor grade and support histopathology, serving as a useful tool for guiding treatments.

In a similar study, HR-MAS was performed on 18 ependymoma, 36 medulloblastoma, and 24 pilocytic astrocytoma tissue samples, and different statistical tests were used to classify them based on their metabolic profiles. HR-MAS data showed that medulloblastomas exhibited significantly higher concentrations of ascorbate, aspartate, PC, taurine, and lipids, as well as lower glucose and scyllo-inositol in comparison to the other tumor types. The researchers in this study then used a Linear Discrete Analysis model to discriminate between the three tumor types and documented the differences in different discriminant functions. They noted that the metabolites that had the potential to separate medulloblastoma included PC, glycine, taurine, and isoleucine, and they could correctly classify medulloblastoma with 94.4% accuracy [38]. They also requested histopathological reports from different online databases, which showed that medulloblastoma had a concordant diagnosis rate of about 90%, the highest of all tumor types studied. Lastly, researchers also utilized a pathway analysis tool, which reported that medulloblastoma had an enriched glycerophospholipid metabolism relative to the other tumor types. This study further indicates that HR-MAS seems capable of accurately detecting tumor grades and that the data it produces can be positively correlated to other methods of diagnosis.

In work performed on overexpressed smoothened (SMO) receptor transgenic mice with high incidence levels of medulloblastoma, HR-MAS was utilized to verify data obtained from ^1^H MRS. The data was positively correlated between the HR-MAS and ^1^H MRS. HR-MAS of four SMO T_2_ lesion-presenting mice in comparison to four wild type and seven normal T_2_ SMO presenting mice, revealed an increase in phosphocholine. Researchers believe this increase accounts for the elevated levels of choline-containing metabolites. The SMO mice exhibited similar metabolic profiles in comparison to human samples that had been previously reported, indicating that they are a promising model for future investigations (Figure 3) [41].

As previously stated, group III and IV mice are characterized by a worse prognosis than other tumor grades. Knowledge surrounding the exact mechanistic pathway is not completely understood, but it has been documented that the MYC gene, a family of proto-oncogenes involved in many cellular processes, is amplified in these cancers. It has been suggested that bromodomain and extraterminal domain (BET) bromodomain inhibitors may be promising for treating MYC-driven medulloblastomas [42]. To confirm, cell lines D283, which can be classified as group III or IV, D458, which can be classified as group III, as well as their respective controls, were treated with a BET inhibitor, OTX-015, for 24 to 48 h. NMR-based metabolomics were then performed, and a metabolic shift that resulted in increased levels of myo-inositol, GPC, UDP-N-acetylglucosamine, glycine, serine, pantothenate, and PC was observed in both cell lines when compared to the control group. An increase in these metabolites is linked to anti-proliferative and cancer cell death. They are also linked to the accumulation of intracellular lipid droplets, which are associated with mitochondrial dysfunction in cancer cells [42]. These results describe molecular and metabolomic differences using NMR metabolomics after treatment with OTX-015, which provides key information for possible mechanisms of action of treatment efficacy. This study also demonstrated the potential of BET inhibitors for the treatment of some MYC-driven medulloblastomas. To further support the findings from this study, the utilization of NMR metabolomics in animal models treated with the BET inhibitor would provide greater insight and relevance in possible clinical translation.

### 2.3. Primary Central Nervous System Lymphoma (PCNSL)

Cerebrospinal fluid (CSF), a product of the choroid plexus, serves many functions and is involved in many biological processes in the brain, such as maintaining interstitial homeostasis, providing protection/acting as a shock absorber, and providing nutrients to the CNS [43]. The utilization of NMR spectroscopy on CSF for metabolomics has proven to be a useful tool for the identification and diagnosis of many brain abnormalities [44]. For example, Primary Central Nervous System Lymphoma (PCNSL) is a rare variant of non-Hodgkin lymphoma (NHL) with a poor prognosis due to its aggressive nature. PCNSL occurs outside of the lymph nodes and begins in parts of the central nervous system such as the brain, spinal cord, orbits, and leptomeninges. Therapies, including high-dose myeloablative therapy alongside autologous stem cell transplantation, high-dose methotrexate-based chemotherapy, and ibrutinib, have been shown to work well for PCNSL despite its poor prognosis [45]. However, for treatment initiation, a brain biopsy must be performed first, which carries a high risk of failure and a high risk of hemorrhage. A 2022 clinical study of PCNSL patients discovered differences in NMR metabolic data of cerebrospinal fluid (CSF) that could be used for PCNSL diagnosis to supplement traditional methods. The study used NMR metabolomics to analyze data on 41 PCNSL patients and 41 normal patients. The findings showed increased lactate, alanine, and citrate, as well as decreased glucose, choline, creatine, malonate, glutamine, and myo-inositol in the PCNSL cohort compared to the normal cohort [46]. Previous findings on brain tumor metabolic data have shown increases in glucose, choline, and glutamine, so the decreases in these metabolites may serve as indicators to differentiate between PCNSL and other brain tumors. Additionally, NMR metabolic data found increased citrate levels in tumors that showed leptomeningeal enhancement on MRI compared with tumors that did not show leptomeningeal enhancement, suggesting metabolic differences between PCNSL tumors on the leptomeninges versus other regions of the brain parenchyma. Additionally, NMR studies on alanine levels in CSF showed a positive correlation with maximum tumor diameter found using contrast-enhanced T_1_WI MRI, indicating potential increased pyruvate to alanine conversion. While previous studies found decreased glucose uptake with high apparent diffusion coefficients (ADC) in brain tumors and lymph nodes, this study found a positive correlation between glucose levels and ADC in the CSF, suggesting that CSF glucose levels may be low in PCNSL patients due to tumor consumption of glucose [46].

### 2.4. Leptomeningeal Carcinomatosis

NMR metabolomics of the CSF has also been used as a diagnostic tool for leptomeningeal carcinomatosis (LC). Leptomeningeal carcinomatosis is the involvement of both solid tumors and hematological cancers in the leptomeninges, the membranes that line the brain and spinal cord. LC is the third most common complication of the central nervous system, with a survival of 1 to 1.5 months in untreated patients and 3 to 6 months with treatment [44]. Diagnosis usually involves the examination of CSF cytologically, but while often successful, it is a method that often lacks sensitivity, so optimization is necessary. Using a designed LC rat model, researchers performed NMR metabolomics on the CSF of 17 LC rats and 7 age-matched control rats. These rats were implanted with GFP-expressing F-98 cells, then 8 of the rats were examined at 3 days post-implantation, while the other 9 rats were examined 7 days post-implantation via cytology, MRI, and NMR. The experimental rats revealed meningeal enhancement, although there was an observed enhancement in rats from the control group as well, indicating a lack of specificity, confirmed via histology. NMR metabolomics was then performed, where it was determined that 7-day post-implanted rats were much different from control groups, with increased lactate, creatine, and acetate levels and decreased glucose levels compared with those of the normal group. With the use of HR-MAS in the LC rat model, late-state diagnosis with this approach demonstrated an 89% overall accuracy of diagnosis [44]. The same research group also published an NMR study using the CSF of 26 LC patients and 41 control patients. This study utilized a solution state rather than the HR-MAS they used in the animal study. The findings showed an increase in lactate, citrate, and alanine in the LC group compared to normal CSF. This was accompanied by decreases in myo-inositol and creatine levels (Figure 4). Although a higher level of creatine was observed in their previous animal study, they attribute the lower level of creatine in the human patients to being a better representation of the actual disease. They also had a greater diagnostic performance of 92% in the human study, which may be due to the larger sample size as well as the greater volume of CSF used in the human studies [47].

### 2.5. Pituitary Tumors

Pituitary tumors, particularly pituitary adenomas, are a diverse group of intracranial neoplasms that often significantly disrupt the endocrine system by altering the secretion of key regulatory hormones such as growth hormone (GH), follicle-stimulating hormone (FSH), luteinizing hormone (LH), adrenocorticotropic hormone (ACTH), and prolactin [48,49]. The disruption in the endocrine system can either detrimentally decrease or increase the production of these hormones, leading to a range of systemic effects, including infertility, acromegaly, Cushing’s disease, and visual disturbances due to mass effect on the optic chiasm. While magnetic resonance imaging (MRI) has proved useful as an imaging modality for detecting pituitary tumors, especially macroadenomas, its results are not fully reliable, as it lacks the sensitivity and specificity required to detect microadenomas or to reliably distinguish malignant tissue from normal pituitary tissue. This limitation can result in both false negatives and false positives, particularly in small or hormonally silent tumors [50].

To address these diagnostic deficiencies, NMR metabolomics can serve as a complementary approach. As discussed earlier in the review, NMR spectroscopy allows for quantitative profiling of metabolites, capturing downstream biochemical changes that reflect altered gene expression, enzymatic activity, and shifts in tumor metabolism—all critical indicators of cancer initiation and progression [51,52]. When applied to pituitary tumor tissue or biofluids (blood, cerebrospinal fluid), metabolomics can detect subtle metabolic shifts that precede tumor formation and progression. Equally important, tumor regression can also be detected after successful treatment. This approach found potential novel biomarkers among distinct tumors. Using ex vivo ^1^H NMR spectroscopy on methanol–chloroform tissue extracts (not HR-MAS or in vivo ^1^H-MRS), prolactinomas exhibited decreased levels of N-acetylaspartate (NAA), myo-inositol, glycine, and taurine, along with increased glutamine compared to gonadotropic (LH/FSH-secreting) tumors. ACTH-secreting tumors showed decreased glycine and phosphoethanolamine, and increased aspartate compared to gonadotropic (LH/FSH-secreting) tumors. The trend of signal neuronal loss suggests that the tumor alters surrounding brain tissue, showcasing that benign pituitary tumors can disrupt the neurochemical environment. Similarly, the increase in aspartate reflects enhanced biosynthetic activity, in turn, possible tumor proliferation due to the overproduction of ACTH. However, the depletion of glycine—a precursor for purines and glutathione—reflects increased nucleotide synthesis. Reduced phosphoethanolamine indicates disruptions in phospholipid metabolism, a common hallmark in proliferating cells [53].

### 2.6. Meningiomas 

Meningiomas are the most common primary intracranial tumors, making up about 30% of all primary CNS tumors. These tumors originate from the arachnoid cap cells of the meninges. Although the majority are classified as WHO grade I and considered histologically benign, their proximity to critical neurovascular structures can result in significant clinical symptoms such as headaches, visual disturbances, and seizures due to mass effect. However, many meningiomas grow slowly, thereby remaining clinically silent or exhibiting non-specific, intermittent symptoms, which complicates early detection and management [54]. Routine MRI has been shown to be relatively effective for localization and monitoring, but provides limited information about tumor grade or biological behavior, especially in atypical presentations. The possibility of abnormal presentations, continued growth, and compromising tumor locations increases the complications in treatment and prognosis. Furthermore, current WHO classification guidelines do not include any molecular or genomic markers. Incorporation of these markers can help improve the diagnosis and stratification of meningiomas, leading to better patient outcomes [55,56].

In recent years, NMR-based metabolomics has emerged as a powerful tool for characterizing the molecular phenotype of tumors through the analysis of biofluids or tissue extracts. This approach is particularly valuable for meningiomas, given the need for non-invasive diagnostic tools and biomarkers predictive of recurrence or malignancy. In a study, a serum metabolomics profiling approach coupled with machine learning algorithms was employed to distinguish meningioma patients from healthy individuals and to stratify tumors by WHO grade [57]. The study identified significant alterations in several metabolites, including decreased levels of GPC, leucine, lysine, and creatine, and increased levels of choline, reflecting dysregulation in lipid metabolism, amino acid biosynthesis, and energy balance. Notably, these metabolic changes correlated with tumor grade, suggesting their potential as prognostic indicators.

The integration of NMR spectroscopy with machine learning classification models enabled high diagnostic accuracy, demonstrating the feasibility of using serum metabolite profiles as non-invasive biomarkers. These findings highlight the potential of metabolomics to support clinical decision-making by offering a metabolic fingerprint of meningiomas, facilitating early detection, grading, and personalized treatment strategies.

### 2.7. Peripheral and Central Nerve Tumors 

Peripheral and central nerve tumors encompass a diverse group of neoplasms, ranging from benign schwannomas and neurofibromas to highly aggressive forms such as malignant peripheral nerve sheath tumors (MPNSTs) and gliomas. These tumors often present diagnostic challenges due to anatomical complexity, non-specific symptoms, and a lack of reliable biomarkers to distinguish malignant from benign phenotypes, especially at early stages [58]. Imaging modalities such as MRI and CT play a central role in localization and structural assessment, but they do not adequately reflect molecular behavior or tumor aggressiveness [59]. As a result, there is growing interest in applying NMR-based metabolomics to uncover tumor-specific metabolic signatures that may support early diagnosis, tumor grading, and treatment stratification. In the context of nerve tumors, such as malignant peripheral nerve sheath tumors (MPNSTs), where histological grading is often subjective and molecular markers are limited, a metabolic profile can help distinguish and diagnose these gradings.

A pivotal study utilized ^1^H NMR spectroscopy to delineate the metabolic profiles of human glioma cell lines, which share molecular characteristics with peripheral nerve tumors. Comparison of low-grade (WHO grade II) astrocytoma-derived cells and high-grade (WHO grade IV) glioblastoma cells revealed malignancy-associated metabolic alterations. High-grade cells demonstrated elevated levels of choline-containing metabolites (e.g., phosphocholine, total choline), indicative of increased membrane synthesis and cell proliferation. Additionally, increased lactate and reduced creatine levels were observed, reflecting enhanced glycolytic activity and disrupted energy metabolism, respectively [60]. Distinct metabolic profiles and characteristics provide critical insights into the molecular mechanisms driving the progression from low-grade, less aggressive tumors to highly malignant phenotypes. However, findings from cell line studies must be further validated in animal models to ensure translational relevance. Such validation can help identify robust imaging biomarkers for noninvasive diagnosis and treatment monitoring.

### 2.8. Dysembryoplastic Neuroepithelial Tumor (DNET)

Dysembryoplastic Neuroepithelial Tumor (DNET) is a benign glioma that has a known link to epilepsy. DNET occurs in the glial and neuronal cells of the cerebellum. DNET has an overall good prognosis, and treatment typically involves surgical resection. NMR spectroscopy has the potential to classify epilepsy-associated neuroepithelial tumors, such as DNET and ganglioglioma, allowing for therapies to be targeted to these diseases when treating epilepsy [61,62].

A clinical study employing HR-MAS NMR spectroscopy compared metabolic data of hippocampi from 5 patients with DNET, 7 patients with gangliogliomas, 20 patients with type 1 hippocampal sclerosis (HS), 6 patients with type 2 hippocampal sclerosis, and 10 patients without hippocampal tumors or any signs of sclerosis. The study found that DNET hippocampi showed higher levels of lactate, alanine, ascorbate, and arginine as well as lower levels of glutathione, acetate, and glutamine when compared with hippocampi without tumors or sclerosis. Additionally, DNET and ganglioglioma showed statistically significant differences in the metabolism of choline. HR-MAS used in this study could potentially allow us to better distinguish varying metabolic pathways between DNET, ganglioglioma, and patients without tumors to improve our understanding of their relationship to epilepsy [62]. Although the sample size is limited, the results are encouraging. Expanding the study with a larger cohort will be an important step toward the clinical translation of these understudied tumor types.

### 2.9. Other Embryonal Tumors (Not Medulloblastoma)

Embryonal tumors are a type of tumor that occurs in the embryonal cells of the brain, which are cells that remain from fetal development [63].

#### 2.9.1. Central Nervous System Atypical Teratoid/Rhabdoid Tumor

Atypical Teratoid/Rhabdoid Tumor (ATRT) of the central nervous system (CNS) is an embryonal tumor with a dismal prognosis due to its aggressive nature. ATRT is one of the most predominant CNS tumors in young children. Treatment of ATRT typically involves conventional chemotherapy, high-dose chemotherapy (HDCT) with autologous stem cell rescue (ASR), and intrathecal (IT) chemotherapy [64].

Although little NMR metabolomics research has been performed on ATRTs, an HR-MAS study was performed on five frozen ATRT human tumor samples in comparison to other tumor types. This study uncovered that in comparison to medulloblastoma, ATRTs had significantly lower creatine. However, given the small sample size used in this study, more work needs to be performed to provide a detailed metabolic profile of ATRT [38].

#### 2.9.2. Embryonal Tumor with Multilayered Rosettes (ETMR)

Embryonal tumor with multilayered rosettes (ETMR) is an embryonal tumor of the brain or spinal canal with alteration of the Chromosome 19 microRNA cluster (C19MC). ETMR is often characterized by large areas of unmyelinated axons, dendrites, and glial cells with rosettes made of neuroepithelial cells [65]. Diagnosis and treatment of ETMR is empirically difficult with a poor prognosis, as it shares many similarities with other primitive neuroectodermal tumors, and its rarity means that there is no standard guideline for treatment. While there have not been reports of NMR metabolomics for ETMR, at least one case of magnetic resonance spectroscopy (MRS) has been performed with small sample sizes, revealing a potential method to improve the diagnosis of the disease [66].

## 3. Future Directions

NMR metabolomics has emerged as a powerful, non-invasive tool for unraveling the complex metabolic alterations associated with a multitude of brain cancers. Its ability to detect and quantify a wide range of metabolites in tissues, biofluids, and serums, in both human, mouse, and cell line models, offers critical insight into tumor biology, progression, and treatment response. NMR’s application in brain cancer research, such as the studies described in this review, has already yielded valuable biomarkers for diagnosis, prognosis, and therapeutic monitoring (Table 1). NMR spectroscopy has the capacity to create metabolic profiles that are patient-specific and can pioneer personalized medicine for brain cancer patients. It has the ability to diagnose cancer type, predict aggressiveness, as well as monitor treatment efficacy. Importantly, the metabolic profiles derived from NMR spectroscopy may also aid in target identification for novel treatments. For example, tumors demonstrating high choline metabolism could be candidates for choline kinase inhibitors, while those reliant on glycolysis could be susceptible to metabolic pathway inhibitors targeting lactate production. The power of NMR metabolomics does not stop with brain cancer patients but expands into any metabolically implicated diseases. Integrating metabolomics with other omics platforms—such as genomics, transcriptomics, and proteomics offers a holistic view of the complex biological dynamics within a system. This multi-layered approach enables researchers to investigate how treatments exert beneficial or adverse effects by identifying metabolite biomarkers and tracking their transformation into epimetabolites that influence physiological regulation [67]. In the context of brain cancer research, combining NMR-based metabolomics with other omics techniques can provide deeper insights into disease mechanisms and therapeutic responses. Such integration holds promise for advancing personalized, point-of-care strategies tailored to individual patients, ultimately improving diagnosis, prognosis, and treatment outcomes.

The application of NMR spectroscopy to obtain rich structural information of molecules has led to significant scientific breakthroughs over the past century. But it is the rapid expansion of computing capabilities in the past decade that has set the stage for the latest revolution of NMR metabolomics, particularly with the utilization of artificial intelligence (AI), machine learning (ML), and deep learning. AI is already changing the current state of NMR metabolomics by enhancing data processing and workflows and enabling faster acquisition times. AI algorithms can automate tasks like spectral interpretation, predict NMR spectra, and even potentially design new NMR experiments. Recent developments in AI-based tools have significantly accelerated NMR data analysis. The DP4-AI platform, designed for automated interpretation of NMR spectra, has demonstrated a 60-fold increase in processing speed compared to manual human analysis [68]. For two-dimensional NMR spectra, tools such as DEEP Picker enable efficient peak picking even in noisy datasets, resolve overlapping peaks, and facilitate accurate peak-to-chemical entity assignments [69].

In the field of metabolomics, AI, machine learning (ML), and deep learning algorithms are increasingly capable of processing complex one-dimensional spectra. These tools can perform baseline correction, metabolite identification, and quantification relative to reference compounds with improved accuracy and reduced error. The future of AI empowered NMR metabolomics in the brain and other organs will set the stage for the next round of innovations in diagnostic and metabolic biomarker space in medicine.

## Figures and Tables

**Figure 1 metabolites-15-00607-f001:**
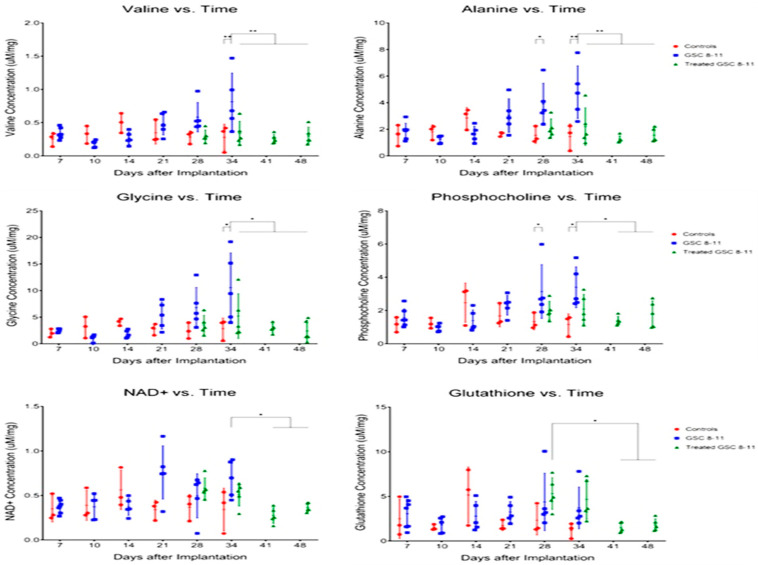
Ex vivo NMR metabolomics of GBM mouse brain tumors found that the individual metabolite pool sizes are significantly altered throughout tumor development and regression following radiotherapy. Statistical significance is indicated by *p* < 0.05 (*) and *p* < 0.01 (**). Adapted from [25].

**Figure 3 metabolites-15-00607-f003:**
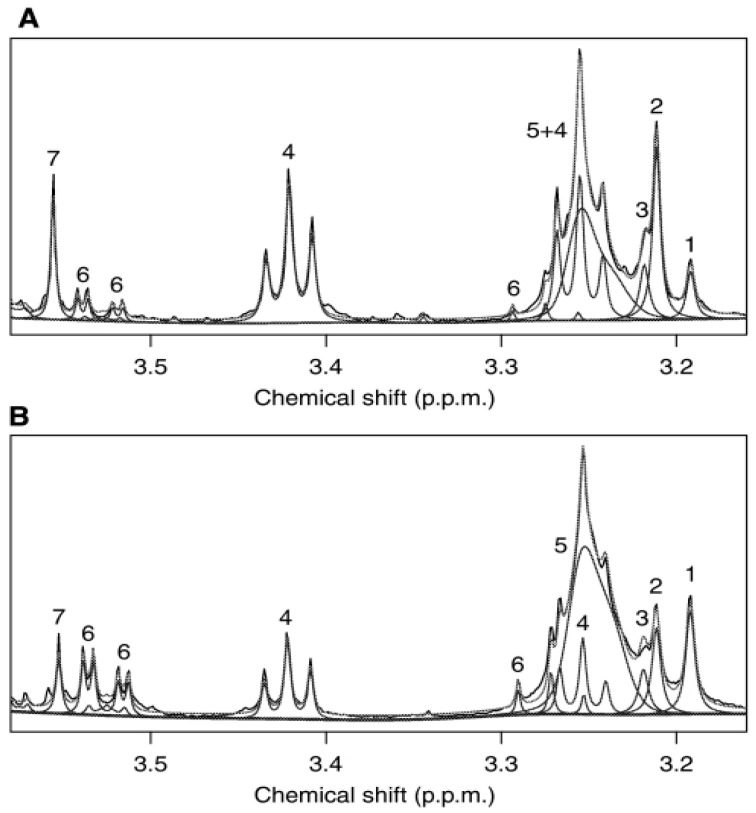
^1^H HR-MAS Spectra (**A**) SMO mouse cerebellum HR-MAS spectra overlaid (**B**) WT mouse cerebellum HR-MAS spectra overlaid. Peaks are (1) Choline, (2) phosphocholine, (3) glycerophosphocholine, (4) taurine, (5) phosphatidylcholine, (6) myo-inositol, and (7) glycine. Figure adapted from [41].

**Figure 4 metabolites-15-00607-f004:**
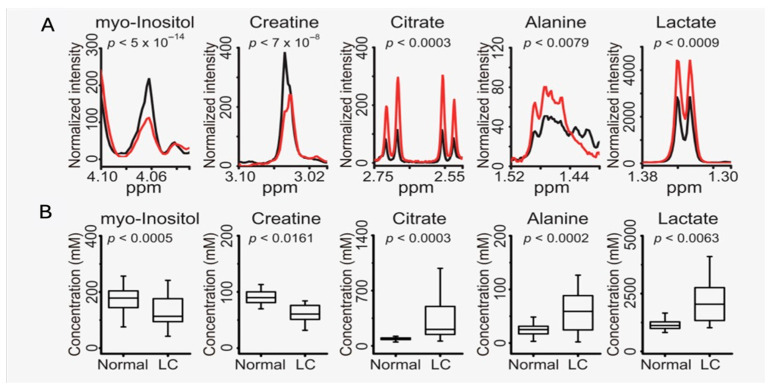
Ex vivo NMR metabolomics of the CSF of rats. (**A**) Significantly changed metabolites between the LC (red) and control (black) groups overlaid. These metabolites include myo-inositol, creatine, citrate, alanine, and lactate. (**B)** The concentration of these metabolites was determined from an internal standard and plotted as a box plot to display concentration differences in these metabolites between groups. Figure adapted from [47].

**Table 1 metabolites-15-00607-t001:** Summary of NMR spectroscopy-based metabolomics study on brain cancer types. Abbreviations include phosphocholine (PC), glycerolphosphocholine (GPC), N-acetyl aspartate (NAA), and myo-inositol (MI).

Cancer Type	Compared to	Sample Used	Increased Metabolites	Decreased Metabolites	References
**Glioblastoma**	Normal	Human Urine and Blood	Phospholipids and total cholesterol		[19]
**Glioblastoma**	Normal	Platelet		Lactate, acetate, glutamine, glutamate, succinate, alanine, and pyruvate	[21]
**Glioblastoma**	Astrocytoma and oligodendroglioma	Human brain tumor		myo-inositol	[28]
**Pilocytic astrocytoma**	Medulloblastomas and ependymomas	Pediatric Tumor Tissue	Leucine, isoleucine, valine, NAA, fatty acids, glutamine, hypotaurine	Choline, PC, GPC, creatine, and MI	[34]
**High-Grade Oligodendroglioma**	Low-grade Oligodendroglioma	Human Tissue	Alanine, valine	Glutamate, glutamine, GABA, NAA, proline	[36]
**Low-Grade Oligodendroglioma**	High-grade oligodendroglioma	Human Tissue	Proline, glutamate, glutamine, GABA, NAA		[36]
**Ependymoma**	Medulloblastoma and Pilocytic astrocytoma	Human tissue	MI and glutathione	GABA, PC, GPC, and leucine	[34,36]
**Medulloblastoma**	Normal	Human Tissue	Taurine, MI, choline, PC, and GPC		[38]
**Medulloblastoma**	Ependymoma, pilocytic astrocytoma	Human tissue	Ascorbate, aspartate, PC, taurine, and lipids	Glucose and scyllo-inositol	[39]
**Primary central nervous system lymphoma**	Normal	Cerebral Spinal Fluid	Lactate, citrate, alanine	choline, creatine, glucose, glutamine, malonate, and MI	[46]
**Leptomeningeal Carcinomatosis**	Normal	Rat cerebral Spinal Fluid	Lactate, creatine, acetate	glucose	[44]
**Leptomeningeal Carcinomatosis**	Normal	Cerebral spinal fluid	Citrate, alanine, and lactate	MI and creatine	[47]
**Prolactinomas**	ACTH-Secreting Tumors	Human Tissue	glumatamine	NAA, MI, glycine, taurine	[51]
**ACTH-Secreting Tumors**	Prolactinomas	Human Tissue	Aspartate	Glycine, phosphoethanolamine	[51]
**Meningiomas**	Normal	Human Serum	Choline	GPC, leucine, lysine, creatine	[57]
**Peripheral and central nerve tumors**	Astrocytoma	Human Cell Lines	Choline, PC, GPC, lactate	Creatine	[60]
**Dysembryoplastic Neuroepithelial Tumor**	Normal	Human Cell Lines	Lactate, alanine, ascorbate, and arginine	Glutathione, acetate, and glutamine	[62]
**Central Nervous System Atypical Teratoid**	Medulloblastoma	Human Tissue		Creatine	[36]

## Data Availability

Not applicable.

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
