# Peer review of "NMR-Based Metabolomic Profiling for Brain Cancer Diagnosis and Treatment Guidance"

_metabolites, 2025, doi:10.3390/metabo15090607_

Round 1

Reviewer 1 Report

Comments and Suggestions for Authors

The review presented by Zickus et al. is a comprehensive summary of the applications of NMR metabolomics in the field of brain cancer as a useful guide for early diagnosis, prognosis and therapy interventions. The review examines exhaustively the different applications of NMR metabolomics in different sub-types of brain cancer.

Only few other reviews on the same topic have been published in the last years (i.e. : Mukherjee et al., Med. Oncol., 2023; Pandey et al., Mol Carcinog., 2017; Salzillo et al.,  Magn. Reson. Imaging Clin. N. Am., 2016) that, maybe, the authors of this paper might cite.

The paper is well written and I believe that it may be interesting for the readers of  the special issue “Harnessing the power of NMR metabolomics in unraveling metabolic diseases” of the journal Metabolites.

Minor points that should be corrected:
Line 16, please correct “NMR specrtoscopy” with “NMR spectroscopy”
Lines 203-204, please insert here the reference [30]
Lines 219-220, please insert also here the reference 
Line 263, please insert also here the reference
Page 14, in Table 1, line 588, 
in Primary central nervous system lymphoma, please correct the wrong reference [40] with the appropriate [42]; in Leptomeningeal carcinomatosis  please correct the wrong reference [38] with the appropriate [40];

Page 15, in Table 1
Please, check if the references cited for Primary centrale nervous system tumors, Dysembryoplastic neuroepithelial tumor and central nervous system atypical teratoid are appropriate 

Author Response

Comment 1: Only few other reviews on the same topic have been published in the last years (i.e. : Mukherjee et al., Med. Oncol., 2023; Pandey et al., Mol Carcinog., 2017; Salzillo et al.,  Magn. Reson. Imaging Clin. N. Am., 2016) that, maybe, the authors of this paper might cite.

Response 1: We thank reviewer for their insights to improve on the review article. The suggested reviews for citation have been included in the revised manuscript in red letters.

Minor points that should be corrected:

Line 16, please correct “NMR specrtoscopy” with “NMR spectroscopy”

Corrected

Lines 203-204, please insert here the reference [30]

Inserted the reference

Lines 219-220, please insert also here the reference

Inserted the reference

Line 263, please insert also here the reference

Inserted the reference

Page 14, in Table 1, line 588,

in Primary central nervous system lymphoma, please correct the wrong reference [40] with the appropriate [42]; in Leptomeningeal carcinomatosis  please correct the wrong reference [38] with the appropriate [40];

We have corrected those typos. The references have been corrected in revised manuscript

Page 15, in Table 1

Please, check if the references cited for Primary centrale nervous system tumors, Dysembryoplastic neuroepithelial tumor and central nervous system atypical teratoid are appropriate

Thank you. We have crosschecked and validated the references for Primary centrale nervous system tumors

Reviewer 2 Report

Comments and Suggestions for Authors

Dear Authors,

I have read your manuscript with great interest. It provides a very comprehensive overview of the applications of NMR-based metabolomics in brain cancer research, spanning multiple tumor types and clinical contexts. The breadth of coverage, clarity in describing metabolic pathways, and emphasis on clinical translation make this review valuable for both researchers and clinicians in the field.

Below, I provide some suggestions that I hope will help strengthen the manuscript further:

Major Points

  1. Organization and readability

    • The review is very detailed, but some sections feel dense. You might consider grouping tumor types into broader categories (e.g., adult gliomas, pediatric tumors, lymphomas and metastatic involvement) to guide the reader more smoothly.

    • A schematic overview comparing common metabolite shifts across cancer types could be very helpful for readers trying to grasp the “big picture.”

  2. Critical perspective

    • Many sections summarize existing studies well, but in places the review could benefit from more critical commentary. For instance, highlighting why certain findings differ between tissue, biofluid, and platelet metabolomics, or discussing the limitations of small sample sizes, would give readers a deeper perspective.

    • Where appropriate, please expand on challenges to clinical translation, such as cost, access to NMR compared with mass spectrometry, or standardization of protocols.

  3. Figures and tables

    • Table 1 is very useful, but the formatting could be refined to improve readability (clearer headings for “upregulated” vs. “downregulated” metabolites, standardized abbreviations).

    • Some figure legends could provide a bit more detail to ensure they stand alone for readers.

  4. Future outlook

    • The section on AI and machine learning is exciting and timely. If possible, please add one or two concrete examples of how these tools are already being applied to NMR spectral analysis or prediction of treatment outcomes.

    • Expanding briefly on the integration of NMR metabolomics with other omics (e.g., genomics, proteomics) would strengthen the case for its role in personalized oncology.

Minor Points

  • Please check grammar and style carefully (e.g., “astrocytoma’s” → “astrocytomas”).

  • Some terms and concepts are repeated (e.g., the role of myo-inositol across tumor types). Condensing these repetitions could streamline the manuscript.

  • Fonts seem to change throughout, kindly make the font universal for the manuscript.

Recommendation: Minor Revision.

This is a well-written and timely review that covers the field in impressive breadth and depth. With some restructuring, additional critical commentary, and minor editorial polishing, the manuscript will make an excellent contribution to Metabolites.

Author Response

I have read your manuscript with great interest. It provides a very comprehensive overview of the applications of NMR-based metabolomics in brain cancer research, spanning multiple tumor types and clinical contexts. The breadth of coverage, clarity in describing metabolic pathways, and emphasis on clinical translation make this review valuable for both researchers and clinicians in the field.

Below, I provide some suggestions that I hope will help strengthen the manuscript further:

ResponseWe sincerely thank the reviewer for their thorough evaluation, attention to detail, and insightful suggestions that have significantly improved the quality of our review. We have carefully addressed all comments and incorporated the recommended changes into the manuscript, which are highlighted in red text.

Major Points

  1. Organization and readability
    • The review is very detailed, but some sections feel dense. You might consider grouping tumor types into broader categories (e.g., adult gliomas, pediatric tumors, lymphomas and metastatic involvement) to guide the reader more smoothly.

                   Response: Thank you for this insightful comment. We have structured the review around the major types of brain                                      cancer to offer readers a clear and concise overview of the current state of research in each subtype.                                             This approach also helps identify areas that require further investigation. Altering the current structure                                         may obscure the intended purpose of the review, which is to highlight both well-studied and                                                           underexplored cancer types.

    • A schematic overview comparing common metabolite shifts across cancer types could be very helpful for readers trying to grasp the “big picture.”

                  Response:  This is an excellent suggestion however we believe this will be repetition of content in Table 1.

  1. Critical perspective
    • Many sections summarize existing studies well, but in places the review could benefit from more critical commentary. For instance, highlighting why certain findings differ between tissue, biofluid, and platelet metabolomics, or discussing the limitations of small sample sizes, would give readers a deeper perspective.

                  Response:  We agree with the reviewer and we have now added the critical insights wherever missing.

    • Where appropriate, please expand on challenges to clinical translation, such as cost, access to NMR compared with mass spectrometry, or standardization of protocols.

                 Response: We have added the section as suggested.

  1. Figures and tables
    • Table 1 is very useful, but the formatting could be refined to improve readability (clearer headings for “upregulated” vs. “downregulated” metabolites, standardized abbreviations).

              Response: Suggested changes are incorporated

    • Some figure legends could provide a bit more detail to ensure they stand alone for readers.

                Response: Suggested changes are incorporated

  1. Future outlook
    • The section on AI and machine learning is exciting and timely. If possible, please add one or two concrete examples of how these tools are already being applied to NMR spectral analysis or prediction of treatment outcomes.

                  Response: We have incorporated the suggestion.

    • Expanding briefly on the integration of NMR metabolomics with other omics (e.g., genomics, proteomics) would strengthen the case for its role in personalized oncology.

                   Response: We have incorporated the suggestion.

Minor Points

  • Please check grammar and style carefully (e.g., “astrocytoma’s” → “astrocytomas”).

      Response: Corrected.

  • Some terms and concepts are repeated (e.g., the role of myo-inositol across tumor types). Condensing these repetitions could streamline the manuscript.

          Response: Incorporated

  • Fonts seem to change throughout, kindly make the font universal for the manuscript.

        Response: Corrected
